# Expression of E4 Protein and HPV Major Capsid Protein (L1) as A Novel Combination in Squamous Intraepithelial Lesions

**DOI:** 10.3390/biomedicines11010225

**Published:** 2023-01-16

**Authors:** Marcin Przybylski, Dominik Pruski, Sonja Millert-Kalińska, Monika Krzyżaniak, Mateusz de Mezer, Magdalena Frydrychowicz, Robert Jach, Jakub Żurawski

**Affiliations:** 1Gynecology Specialised Practise, 60-682 Poznań, Poland; 2Department of Obstetrics and Gynecology, District Public Hospital in Poznan, 60-479 Poznań, Poland; 3Gynecology Specialised Practise, 60-408 Poznań, Poland; 4Doctoral School, Poznan University of Medical Sciences, 61-701 Poznań, Poland; 5Department of Pathology, Hospital of Lord’s Transfiguration, Poznan University of Medical Sciences, 61-848 Poznań, Poland; 6Department of Immunobiology, Poznan University of Medical Sciences, 60-806 Poznań, Poland; 7Department of Gynecological Endocrinology, Jagiellonian University Medical College, 31-008 Cracow, Poland

**Keywords:** E4, L1, HPV major capsid protein, LSIL, HSIL

## Abstract

We aim to describe the relationship between the immunohistochemical expression patterns of HPV E4 markers and the presence of HPV major capsid protein (L1) in cervical tissues obtained by biopsy of patients with abnormal liquid-based cytology (LBC) results, HR HPV infections, or clinically suspicious cervix. A novel HPV-encoded marker, SILgrade-E4 (XR-E4-1), and an HPV (clone K1H8) antibody were used to demonstrate the expression in terminally differentiated epithelial cells with a productive HPV infection in the material. A semiquantitative analysis was performed based on light microscope images. The level of E4 protein decreased with the disease severity. Patients with LSIL-CIN 1 and HSIL-CIN 2 diagnoses had significantly lower levels of HPV major capsid protein (L1) than those without confirmed cervical lesions. Our analysis confirms a higher incidence of L1 in patients with molecularly diagnosed HPV infections and excluded lesions of LSIL-CIN 1 and HSIL-CIN 2. Further studies on the novel biomarkers might help assess the chances of the remission of lesions such as LSIL-CIN 1 and HSIL-CIN 2. Higher levels of E4 protein and L1 may confirm a greater probability of the remission of lesions and incidental infections. In the cytological verification or HPV-dependent screening model, testing for E4 protein and L1 expression may indicate a group with a lower risk of progression of histopathologically diagnosed lesions.

## 1. Introduction

The human papillomaviruses (HPVs) are a heterogeneous family of viruses with over a hundred known subtypes [1]. Professor zur Hausen, awarded the Nobel Prize, made an undeniable contribution to the description and understanding of human papillomaviruses’ role in the pathogenesis of cervical lesions [2]. Since then, this group of viruses was carefully studied. It is worth noting that approximately 90% of the population is infected with HPV at some point in their lifetime, and according to CDC, more than 42 million Americans are currently infected with HPV types that cause disease, and about 13 million Americans, including teens, become infected each year [3].

HPV is characterised by tropism to the multilayered epithelium of the skin and mucous membranes. As a result of microdamage to the skin or epidermis, the cells of the basal layer are infected [4]. The replication and multiplication of HPV in the human body are strictly conditioned by viral and cellular regulatory proteins and related to the process of epithelial cell differentiation [5,6].

The clinical outcome of infection varies depending on the type of HPV. Human papillomaviruses are responsible for forming warts, condylomas, precancerous skin and mucous membrane conditions, intraepithelial neoplasia, squamous cell carcinoma, and adenocarcinoma of the cervix. The HPV family of viruses is heterogeneous and can be divided into *α*, *β*, *γ*, *μ*, and *ν*. A group of HPVs that are genitally transmitted comprises supergroup A (known as *α*-papillomaviruses) [5], and we distinguish them as both benign and malignant. These include subtypes such as HPV 6 and 11, which have a low oncogenic potential but are responsible for forming wart-like lesions, which may present both in the genito-anal area and the oral cavity. By contrast, persistent HPV infection with high-risk genotypes, such as 16, 18, 31, 33, 45, and others, is a direct and undeniable factor in squamous and glandular intraepithelial lesions and cervical cancer. Supergroup B is responsible for skin infections, while genotypes from the γ, μ, and ν groups frequently lead to the formation of warts that do not undergo neoplastic transformation. Due to the multitude and variety of lesions that human papillomavirus infections may induce, prevention is essential, including vaccination against HPV and the detection and treatment of early lesions. Important information during population vaccination is the prevalence of particular HPV genotypes in a given area. 

The microscopic evaluation of tissue preparations obtained from a cervical biopsy or loop electrosurgical excision procedure (LEEP) by experienced pathologists is crucial for diagnosing precancerous lesions. There are clearly defined differences and guidelines for distinguishing between LSIL-CIN 1, HSIL-CIN 2, and HSIL-CIN 3 lesions; it is a subjective assessment. Researchers prove that, especially in the case of HSIL-CIN 2 diagnosis, it is easy to make mistakes—the aspects of underestimation and overtreatment are essential.

The use of p16 IHC staining enables classifying CIN 2 lesions as the LSIL or HSIL type. It could limit the number of surgical procedures, such as LEEP conisation or cold knife conisation [7]. Due to the above-mentioned diagnostic difficulties and, thus, a separate therapeutic procedure, objective methods of tissue evaluation are sought. In addition, biomarkers that might unequivocally help to make the final histopathological diagnosis are desired. Immunohistochemical staining, such as p16 (alone or in combination) and Ki67, is well established. The virus’ replication is correlated with the host cell differentiation stages. According to Doorbar et al., it is a critical determinant for effective viral replication and potentially crucial in virus–host immune interaction [6]. HPV particles are in the capsid form, and their greatest accumulation in the superficial layers is related to the epithelial development cycle. We distinguish the episomal form and the integrated form of the virus. During natural infection, HPV produces a viral protein to facilitate replication and production. The E4 protein belongs to the group of regulatory proteins encoded by the early virus genes [8]. The function of the E1, E2, E4, E5, E6, and E7 proteins is to maintain the viral genome in infected cells and to replicate. The E4 protein is one of the less understood proteins; it is probably involved in the release of daughter virions from the cell in its upper layers by destabilising keratin fibres and stopping the cell cycle in the G2 phase. E4 protein is expressed in the superficial layers slightly earlier than the L1 protein [9,10].

The E4 protein’s function is thought to induce cell cycle arrest and disrupt the keratin filaments [11]. It may also facilitate efficient viral release and transmission. As the E4 protein is deposited as amyloid fibres, it can be used as a biomarker of active virus infection and disease severity [12,13]. The two late genes encode the L1 and L2 proteins, which form an icosahedral capsid around the HPV genome. The L1 protein has DNA-binding activity. During infectious entry, the nonenveloped virion uncoats in the endosome; after that, conformational changes result in the dissociation of L1 from L2, which remains in complex with the HPV DNA. The capsid proteins L1 and L2 are critical for virion assembly [14]. Both capsid proteins are essential in interactions with cellular macromolecules that facilitate viral entry into keratinocytes. A large number of HPV major capsid protein (L1) confirms the strong replication of the virus in cells without squamous intraepithelial lesions and may indicate an early phase of increased replication. The most conserved fragment in the HPV genome is the region encoding the E1 and L1 proteins [5]. The HPV types and genotypes are distinguished based on at least a 10% difference within the L1 gene sequence. Isolates of a virus type whose L1 genes differ from an established type by 2–10% and intermediates between types and variants are considered subtypes.

The present study aims to characterise the relationship between the immunohistochemical expression patterns of HPV E4 markers and the presence of HPV major capsid protein (L1) in cervical tissue obtained by biopsy. Additionally, we would like to assess their relationship with HPV genotypes in LBC from women referred for colposcopy due to abnormalities in previous punch biopsies. Moreover, we examine the relationship between E4 protein and the status of HPV infection. Another research question is whether the genotyped subtypes are related to the presence of HPV major capsid protein (L1) in the tissue. All the examination and follow-up groups are under regular oncogynaecological care and underwent proper treatment.

## 2. Materials and Methods

### 2.1. Study Design

We provide a prospective, ongoing 24-month, non-randomised pilot study to assess the level of HPV E4 protein and the presence of HPV major capsid protein (L1) in patients reporting to the District Hospital in Poznań and Specialised Individual Practise due to abnormal LBC results, the presence of highly oncogenic HPV genotypes, or clinically suspicious cervix in the years 2019–2021. The Poznan University of Medical Sciences Bioethical Committee approved the study protocol (540/22). We obtained written consent for the study from all the patients. We included patients who met the following criteria: (i) aged over 18; (ii) non-pregnant subjects, at least six weeks after the puerperium for patients who gave birth; (iii) patients not treated with immunosuppressive drugs; (iv) expressing informed and written consent to participate in the study; (v) agreeing to the proposed surgical diagnostics in the case of indications and possible surgical treatment. The exclusion criteria were (i) the refusal of the possible treatment of SIL and (ii) a lack of technical possibility of performing the test. A total of 85 women met the above criteria. 

All the subjects from the study group underwent a verification diagnostic analysis of abnormal LBC results. Either punch biopsy or, in the case of histopathologically confirmed HSIL (CIN 2 and CIN 3), LEEP-conisation and curettage of the cervical canal was performed.

### 2.2. Specimen Collection and Handling

#### 2.2.1. LBC and HPV Genotyping Test

We collected liquid-based cytology (LBC) and molecular assessment samples with an endocervical cyto-brush preserved in PreservCyt^®^ (Hologic Corp, Marlborough, MA, USA). Then, the probes were passed to an independent, standardised laboratory. Cervical swabs were analysed according the Bethesda system. PCR was performed, followed by a DNA enzyme immunoassay and genotyping with a reverse hybridisation line probe assay for HPV detection. The lab technicians performed sequence analysis to characterise HPV-positive samples with unknown HPV genotypes. The molecular test detected the DNA of 41 HPV genotypes to identify the following genotypes: 6, 11, 16, 18, 26, 31, 33, 35, 39, 40, 42, 43, 44, 45, 51, 52, 53, 54, 55, 56, 58, 59, 61, 62, 64, 66, 67, 68, 68a, 68b, 69, 70, 71, 72, 73, 81, 82, 83, 84, 87, CP6108, and 90 in vitro. A positive result in molecular tests confirms the presence of DNA of at least one of the mentioned above oncogenic types of HPV in the collected specimens. 

The LINEAR ARRAY HPV Genotyping Test uses biotinylated primers defined to nucleotide sequences within the L1 region of the HPV genome. The HPV primer pool in the master mix reagent is used to amplify the DNA of 41 genotypes of the virus. Within the L1 regions, bound by the primers, sequences connect with the probes. An additional pair of primers targets the human beta-globin gene to ensure the adequacy of the cellular material, extraction, and amplification of the genetic material. AmpliTaqR Gold DNA Polymerase is used for “hot start” amplification of the HPV target DNA and the control beta-globin gene. After PCR amplification, the HPV amplicon and beta-globin amplicon are denatured with a denaturation solution to create single-stranded DNA. Sequential volumes of the amplicon are transferred to holes containing a hybridisation buffer and a single LINEAR ARRAY HPV genotyping strip (HPV and beta-globin-specific probes are present on the strip). Only an amplicon containing highly matched sequences (only 1–3 mismatches) with the probe can hybridise to it. After hybridisation, streptavidin–horseradish peroxidase conjugate is applied to the LINEAR ARRAY HPV genotyping strip, followed by hydrogen peroxide and TMB. The reaction produces a blue-coloured complex that shows up at the probe position where hybridisation occurred. Then, the technicians perform a visual analysis by comparing the blue line pattern to the LINEAR ARRAY HPV genotyping test reference guide result template.

#### 2.2.2. Colposcopy and Punch Biopsy

The Polish Society of Colposcopy and Cervical Pathophysiology recommended the classification of the International Federation of Cervical Pathology and Colposcopy. Each time, we performed a punch biopsy from clinically suspect sites and curettage of the endocervix [15,16]. 

### 2.3. Immunohistochemistry

To test the expression of HPV and E4 markers, 30 slides were stained using immunohistochemistry methods. They contained tissue samples from 85 patients. On each slide were samples from six patients assessed by two independent pathologists. A pathology specialist identified and marked areas of tissue containing lesions for the construction of the slides. Patients’ blocks were assembled using a UNITMA Quick-Ray^®^ Manual Tissue Microarrayer. Specimens were retrieved from selected regions of donor tissue and precisely arrayed in a new recipient paraffin block. The tissue cores were 5.0 mm in diameter and ranged in length from 1.0 to 6.0 mm, depending on the depth of tissue samples available in the donor block. Cores were inserted into 14 × 14 × 5 mm recipient blocks. The initial sections were stained for haematoxylin and eosin to verify the histopathological findings. 

The presence of HPV in cells was detected using a human papillomavirus (HPV) Ab-3 (Clone K1H8) antibody (Dianova GmbH, Hamburg, Germany, Cat. #DLN-14562). E4 expression was detected using a SILgrade-E4 (XR-E4-1) (Labo Bio-medical Products BV, Rijswijk, The Netherlands, REF K-162-C). The expression levels of both proteins were evaluated based on visualisation by immunohistochemical staining, described below.

#### 2.3.1. Immunohistochemistry: HPV Major Capsid Protein (L1) and E4 Protein

Serial 4-micrometre tissue sections were cut from the donor blocks containing cores of lesions and applied to adhesion slides (Epredia™ SuperFrost Plus™). A human papillomavirus (clone K1H8) antibody (Dianova GmbH, Hamburg, Germany) was used to visualise cells infected with HPV types 6, 11, 16, 18, 31, 33, 42, 51, 52, 56, and 58. The HPV major capsid protein (L1) in the tissues were detected using an antibody to the nonconfirmational internal linear epitope of the major capsid protein of HPV 6, 11, 16, 18, 31, 33, 42, 51, 52, 56, and 58. The specification from Dianova is enclosed. 

A novel HPV-encoded marker, SILgrade-E4 (XR-E4-1) (DDL Diagnostic Laboratory, Rijswick, Netherlands), was used to demonstrate expression in terminally differentiated epithelial cells with productive HPV infection in the tissue material. 

Slides were stained on a fully automated immunohistochemistry slide stainer, BenchMark ULTRA (Ventana Roche, Oro Valley, AZ, USA). The staining protocol parameters were based on HIER using CC1 (a heating time of 24 min, at 100 °C), protease 3 (760-2020) for 4 min, 32 min of incubation with the primary Ab, and OptiView (760-700) with amplification (760-099) as a detection system. The antigen was localised using chromogen DAB-3.3 applied in all the preparations. The slides were stained with hematoxylin II (790-2208) for 8 min and bluing reagent (760-2037) as a post counterstain for 4 min. The slides were passed through a series of alcohols and, finally, xylene before the coverslips were mounted. Before staining our study group, we prepared slides with the HPV-infected cervix as a positive control and the tonsil as a negative control.

#### 2.3.2. Light Microscopy Techniques for Cell Imaging

The slides were photographically documented using a 3DHISTECH pannoramic MIDI scanner (3DHISTECH, Budapest, Hungary, calibrated with cellSence software). The magnification was set at 400×. The staining was evaluated semiquantitatively using an Olympus BX 43 light microscope and the cellSens Dimension software from Olympus. Automatic detection of objects with a colour intensity greater by limit value enabled the identification of areas with a positive immunohistochemical reaction occurrence. The software detected the coloured area in the required intensity and recorded it as a numerical value in mm^2^ in the MS Exel file. The analyses included photos of the examined tissue fragments from patients who showed symptoms of HPV infection. DAB (brown chromogen, 3,3′-diaminobenzidine) staining was quantified by phase analysis. The brown-coloured staining was indicative of the expression of the analysed protein. Automatic classification and measurements were performed and used for further statistical analysis [17]. 

Figure 1 presents an immunohistochemical analysis of the expression of L1 and E4 proteins of HPV in epithelial cells in the tissue of two sample patients. The tissue from patient A was histopathologically determined as LSIL-CIN 1, while the genotype of the HPV was positive for 33 and 62. In turn, the tissue of patient B belongs histopathologically to HSIL-CIN 2, and the genotype of the HPV was positive for 52.

### 2.4. Statistical Analysis

We conducted the analysis in the statistical program SPSS and set the *p* value as 0.05. Nominal variables are presented as n and %; quantitative ones, as the median (Me) with quartiles 1 and 3; and the median difference (MD) is specified with 95% confidence intervals. The normality of the variables’ distributions was analysed with Shapiro–Wilk tests. All the tests that were used in the analysis were nonparametric; to compare the level of E4 protein and the level of HPV antibodies, the Mann–Whitney U test or Kruskal–Wallis test was conducted (depending on the number of groups). Kendall’s tau-b was used as a correlation coefficient to analyse the dependency between two quantitative variables. Boxplots were constructed with the Python programming language using the Plotly Graphing Library.

## 3. Results

As shown in Table 1, most patients were diagnosed with LSIL-CIN 1 (44.7%). Almost one-third of the women received HSIL-CIN 2 diagnoses from the cervical biopsies, whereas one-fifth had HSIL-CIN 3 diagnoses. Only four women had no pathology in the histopathological results. One-third of the study group had comorbidities. The most frequent diseases were hypothyroidism, Hashimoto’s disease, insulin resistance, polycystic ovary syndrome (PCOS), and infertility. The mean level of the E4 protein (expressed as the surface area of the immunochemical signal) reached 108.02 µm^2^, and the average level of HPV major capsid protein (L1) reached 5008.66 µm^2^. Regarding the LBC results, most of the patients were diagnosed with LSIL (43.5%). A similar percentage of patients had the following LBC results: ASC-US, ASC-H, and HSIL. Only three patients had no pathology found in the LBC. In one patient, the cervical swab turned out to be non-diagnostic. Therefore, we did not use these data in the analysis. A significant majority of the women were HPV-positive (88.2%); the most frequent genotypes were HPV 16 (positive in over half of the study group), HPV 31, and HPV 6. Other HPV genotypes were present in less than 10% of the study group. 

Subjects with the HPV 18 genotype had significantly higher levels of E4 protein than subjects without this genotype of HPV (MD = −3273.52; 95% CI = −8990.23; −138.01; and *p* = 0.017). Women with the HPV 6 genotype had lower E4 protein levels than women without it (MD = 103.06; 95% CI = 1.91; 1106.53; and *p* = 0.036). No other significant differences in the level of E4 protein between the selected groups were observed (histopathological outcome, HPV (−), other HPV genotypes, and comorbidities), as presented in Table 2.

Moreover, it should be noted that the level of E4 protein decreased with increasing intraepithelial lesions; it reached 226, 120, and 45 for CIN 1, CIN 2, and CIN 3, respectively. Patients from the CIN 3 group had significantly lower levels of E4 protein than patients with CIN 1 and 2 analysed as one group (MD 95% CI = −170.63 (−426.55; −3.69); *p* = 0.024). 

When analysing all four groups with outliers, patients with CIN 1 and CIN 2 diagnoses had significantly lower levels of HPV major capsid protein (L1) than patients whose biopsy results were normal (*p* = 0.002 for mail analysis; *p* < 0.050 for both post hoc analyses). After excluding the “norm” group from the analysis, patients with CIN 3 diagnoses also significantly differed from the CIN 1 and CIN 2 groups—the level of L1 was significantly higher in this group, both before and after removing extreme outliers from the CIN 1 group (*p* = 0.022 for the main analysis with outliers, and *p* = 0.015 for the main analysis without outliers *p* < 0.050 for all the post hoc analyses). Patients with CIN 3 diagnoses also had significantly higher levels of L1 when the CIN 1 and CIN 2 groups were analysed as one group (MD 95% CI = 11598.51 (949.47; 14437.27); *p* = 0.006). Moreover, the patients with normal biopsy results had significantly higher levels of a dependent variable than patients from the CIN 2 group (*p* = 0.001 for main analysis; *p* < 0.050 for all the post hoc analyses), as shown in Table 3. Figure 2 and Figure 3 present the outcomes of the analysis. 

No correlation between the HPV major capsid protein (L1) and E4 protein level was detected (tau-b = 0.004; *p* = 0.952), as shown in Figure 4.

## 4. Discussion

Our study aimed to describe the relationship between the immunohistochemical expression patterns of HPV E4 markers and the presence of HPV major capsid protein (L1) in cervical tissue obtained by biopsy. Additionally, we assessed their relationship with specific HPV genotypes found in patients. Our study confirmed that subjects with the HPV 18 genotype had significantly higher E4 protein levels than subjects with a different genotype of HPV. By contrast, women with HPV 6 had lower E4 protein levels. Interestingly, for the HPV 16 genotype, the *p*-value reached 0.056. It is also worth noting that we considered a p significance level of <0.05; however, possibly upon enlarging the study group, the results might become statistically significant. 

As expected, we observed that the level of E4 protein decreased with an increase in intraepithelial lesions; it reached 226, 120, and 45 for CIN 1, CIN 2, and CIN 3, respectively. These results are in line with Leehman’s work, which confirmed a decreasing amount of E4-positive staining from 41% in LSIL to 3% in HSIL lesions (CIN 3). In addition, Leehman’s work touches on using another immunohistochemical staining, e.g., p16. The spread of p16 is closely related to the epithelial development cycle, and its expression is related to the production and transformation of HPV. Additionally, it has to be emphasized that the intensity of p16 expression and the spread to higher layers of the epithelium is correlated with the disease severity [18]. 

The research by Vink et al. also confirms low E4 protein levels in HSIL/CIN3 cervical lesions. E4 protein expression was present in 9.8% of CIN 3 [13].

The HPV-encoded marker panE4 might be considered a novel marker for the initiation of the viral productive phase and, hence, the completion of the papillomavirus life cycle [12]. It is expressed in productive HPV infection in differentiated, mature epithelial cells [19,20,21]. HSIL/CIN3 is almost always negative for E4, while HSIL/CIN2 and LSIL/CIN1 may be either E4-positive or negative [19,21]. The current SIL/CIN classifications do not distinguish markers corresponding to productive or transforming infections. Observing the patterns of expression of biomarkers such as E4 might play a crucial role in predicting the progression of a lesion. In the future, it may reduce the overtreatment of productive lesions that can regress [18]. 

More publications support the idea that extensive E4 protein expression decreases with an increasing CIN grade. According to Zummeren et al., E4 protein expression is most frequent in classically graded CIN 1 and absent in carcinomas [22]. High-grade lesions (CIN 2/3) showed less E4 expression, which was inversely related to increasing hypermethylation. Promoter hypermethylation of host-cell genes involved in cervical carcinogenesis is a marker for an advanced transforming HPV infection. During HPV-induced cervical carcinogenesis, the methylation levels increase with the severity of the underlying cervical disease and are exceptionally high in cervical cancer. The findings of Zummeren et al. illustrate the gradual transition of productive CIN (reflected by extensive E4 expression) to advanced transforming CIN (reflected by extensive hypermethylation) and cancer. The slightly higher prevalence of anti-HPV-16 E4 antibodies in cervical cancer patients, as initially observed by Jochmus-Kudielka et al. [23] and confirmed later [24], might be explained either by the continuous expression of the E4 protein in some cases of cervical cancer, even in the absence of virus production, or by a concurrent presence of virus-producing benign intraepithelial lesions in those patients [25,26]. An alternative carcinogenic pathway may be characterised by E2/E4/E5 expression. Half of the HPV-positive cervical cancers comprise a subtype with an increase in the expression of E2/E4/E5 and an association with a lack of integration into the host genome [27].

The second aspect of our work was to assess the presence of HPV major capsid protein (L1) in cervical tissues, which, to date, is not explored in the literature. Patients with CIN 1 and CIN 2 diagnoses had significantly lower levels of L1 than patients with normal histopathological results. Our analysis confirms a higher incidence of HPV in patients with molecularly diagnosed HPV infections with excluded cervical lesions of LSIL-CIN 1, HSIL-CIN 2, and CIN 3. For this reason, the above results exclude the use of the test detecting the HPV major capsid protein (L1) in cervical tissues for a more accurate detection of groups at high risk of developing lesions such as HSIL and cervical cancer. On the other hand, such a test may help in detecting incidental HPV infections. As Hilfrich et al. proves, the L1 capsid protein is detectable only at that stage of the life cycle. Therefore, detecting L1 capsid protein synthesized in the cells of the superficial layer of the epithelium might be easy to obtain while taking a cervical swab [28]. However, diagnosis of the above-mentioned infections should be followed by the collection of histopathological material from the cervix for IHC analysis. 

We believe that it would be valuable for science and beneficial for patients in the future to develop diagnostic tests detecting L1 and E4 protein in cervical smears, which would significantly reduce the invasiveness of the test. However, it should be borne in mind that the cervical smear is superficial and we could obtain few L1 or E4-containing cells, while in tissue sections, the assessment of layers and individual cells is much easier due to the amount of material.

A small group of studies indicate the possibility of using an immunohistochemical test detecting HPV major capsid protein (L1) and the E4 protein in a population of HPV-positive patients to identify incidental HPV infections with excluded cervical intraepithelial lesions. In the future, they may also facilitate the diagnostic and therapeutic process for intraepithelial lesions in the cervix and the vagina, vulva, anus, and nasopharynx. 

Interestingly, in tissues with confirmed pathology, observed highest levels of L1 in CIN 3 lesions might be associated with the development of THIN HSIL. THIN HSIL lesions characterized by faster development within the metaplastic immature epithelium or glandular epithelium with rapid proliferation of reserve cells may be characterized by high expression of L1. Alternatively, coincidental infection with a new oncogenic type of HPV virus during the development of a CIN 3 lesion and the patient’s current immune status could impact the level of replication, and thus expression of L1.

In the cytological verification or HPV-dependent screening model, testing for E4 or L1 expression may indicate a group with lower risk of progression of histopathologically diagnosed lesions of CIN 1 and 2 type. Additionally, it could extend the observation time of CIN 1 and CIN 2 without the need for surgical intervention (e.g., LEEP-conisation).

## 5. Conclusions

A significantly higher diagnostic value characterizes E4 protein expression analysis compared to the L1 expression analysis. The study showed only the fact that the level of the expression of the E4 protein decreases with the disease severity. Finally, it may be a marker stratifying women with histopathological diagnosed cervical intraepithelial neoplasia into a progressive or non-progressive group, but further analysis will be needed to confirm this thesis. The final decision, regarding a surgical diagnosis, is always made by experienced physicians using an adequate colposcopic protocol.

## Figures and Tables

**Figure 1 biomedicines-11-00225-f001:**
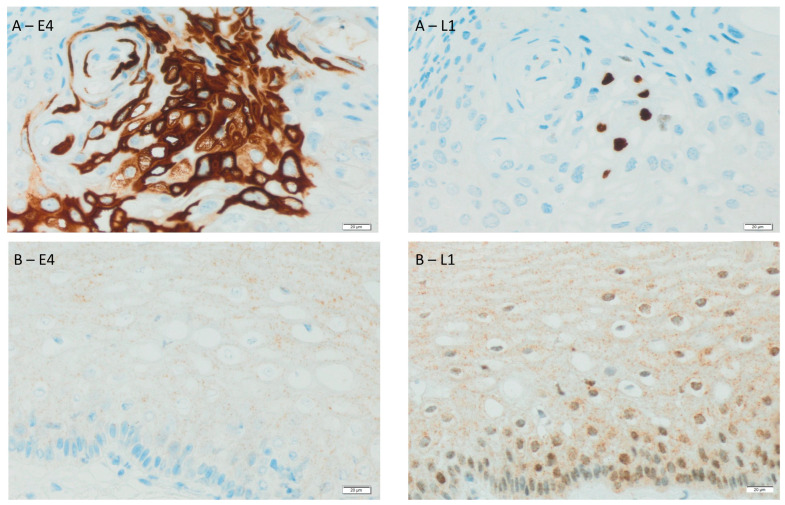
Immunohistochemical expression of HPV in epithelial cells in the tissue of the same patients (**A**,**B**). Detection was observed using antibody Ab-3 (clone K1H8) detected major HPV capsid protein L1 and E4 SILgrade-E4 (XR-E4-1) detected E4, a protein marker for initiation of a productive phase of the HPV life cycle. The brown colour indicates an immunopositive immunohistochemical reaction for the tested markers. Magnification 400×.

**Figure 2 biomedicines-11-00225-f002:**
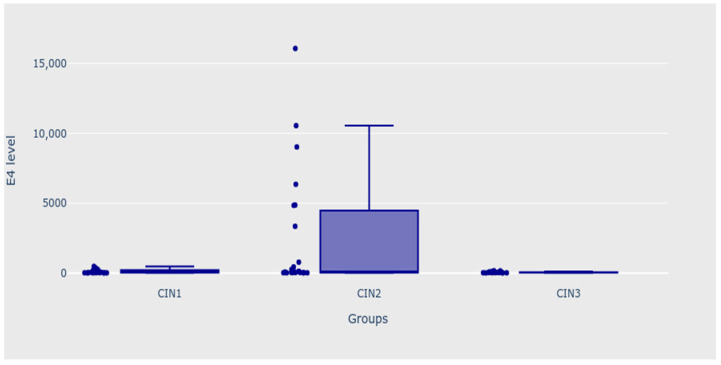
Levels of E4 protein—CIN 1, CIN 2, and CIN 3 groups.

**Figure 3 biomedicines-11-00225-f003:**
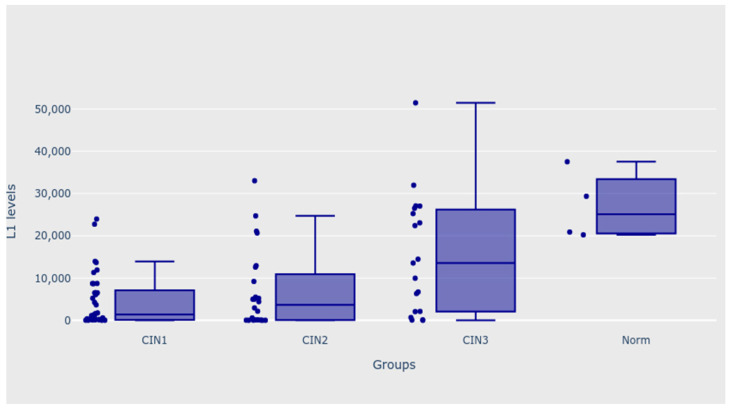
Levels of L1—CIN 1, CIN 2, CIN 3, and norm groups.

**Figure 4 biomedicines-11-00225-f004:**
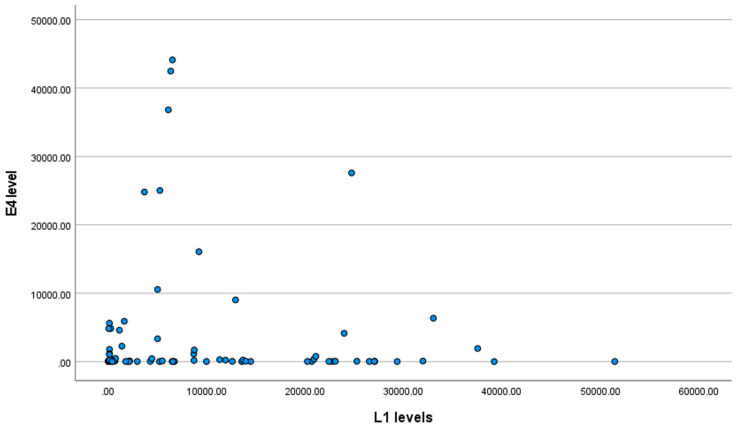
Scatter plot for the E4 protein level and L1 level.

**Table 1 biomedicines-11-00225-t001:** Characteristics of the study group including the age of patients, level of cervical intraepithelial neoplasia, results of LBC, and HPV status with the genotype of the virus.

Characteristic	Value
Mean age	32.00
Histological outcome	
CIN 1	38 (44.7%)
CIN 2	24 (28.2%)
CIN 3	19 (22.4%)
No pathology	4 (4.7%)
E4 protein level, Me (Q1; Q3)	108.02 (35.60; 1712.73)
HPV L1 level, Me (Q1; Q3)	5008.66 (154.65; 13,688.40)
LBC result, n (%)	
NILM	3 (3.5%)
ASC-US	15 (17.6%)
ASC-H	16 (18.8%)
LSIL	37 (43.5%)
HSIL	13 (15.3%)
None	1 (1.2%)
HPV status	
PV (+)	75 (88.2%)
PV (−)	10 (11.8%)
HPV genotype, n (%)	
16	40 (53.3%)
18	5 (6.7%)
31	12 (16.0%)
45	4 (5.3%)
6	9 (12.0%)
Other genotypes	5 (6.7%)

CIN—cervical intraepithelial neoplasia; HPV—human papillomavirus; LBC—liquid-based cytology; NILM—negative for intraepithelial lesion or malignancy; ASC-US—atypical squamous cells of undetermined significance; ASC-H—atypical squamous cells—cannot exclude HSIL; LSIL—low-grade squamous intraepithelial lesion; and HSIL—high-grade squamous intraepithelial lesion.

**Table 2 biomedicines-11-00225-t002:** Comparison of E4 protein level between selected groups.

Variable	E4 Protein Level	MD (95% CI)	*p*
Histological outcome (with outliers)			
CIN 1 (n = 38)	226.89 (36.83; 1795.09)		0.156
CIN 2 (n = 24)	120.95 (39.48; 4857.83)		
CIN 3 (n = 19)	45.06 (31.68; 102.28)		
No pathology (n = 4)	200.72 (29.70; 1147; 08)		
Histological outcome (with outliers)			
CIN 1 (n = 38)	226.89 (36.83; 1795.09)		0.079
CIN 2 (n = 24) CIN 3 (n = 19)	120.95 (39.48; 4857.83) 45.06 (31.68; 102.28)		
Histological outcome (with outliers)			
CIN 1 + CIN 2 (n = 62)	215.69 (37.37; 3348.77)	−170.63 (−426.55; −3.69)	0.024
CIN 3 (n = 19)	45.06 (31.68; 102.28)		
HPV status			
Negative (n = 10)	70.23 (35.18; 111.75)	59.93 (−21.23; 783.87)	0.354
Positive (n = 75)	130.16 (36.12; 2029.93)		
HPV genotypes			
16			
Yes (n = 40)	57.67 (32.42; 388.17)	150.49 (−0.59; 312.10)	0.056
No (n = 45)	208.16 (38.22; 4151.30)		
18			
Yes (n = 5)	3348.77 (2254.76; 9026.87)	−3273.52 (−8990.23; −138.01)	0.017
No (n = 80)	75.25 (35.37; 1084.38)		
31			
Yes (n = 12)	1262.60 (76.26; 15688.67)	−1188.83 (−4843.74; 4.19)	0.067
No (n = 73)	73.77 (35.18; 1045.23)		
45			
Yes (n = 4)	1141.35 (953.21; 5093.02)	−1064.62 (−4180.16; 671.57)	0.062
No (n = 81)	76.73 (35.56; 1712.73)		
6			
Yes (n = 9)	41.59 (25.22; 66.89)	103.06 (1.91; 1106.53)	0.036
No (n = 76)	144.65 (36.22; 2097.55)		
More than one HPV genotype			
Yes (37)	260.44 (33.80; 4850.36)	−185.19 (−1121.81; 14.02)	0.321
No (48)	75.25 (37.49; 335.01)		
Comorbidities			
Yes (26)	108.02 (32.32; 3348.77)	−18.60 (−220.85; 42.94)	0.934
No (59)	89.42 (36.74; 1141.35)		

CIN—cervical intraepithelial neoplasia; HPV—human papillomavirus. MD 95% CI—the median difference with the 95% confidence interval. Comparisons were made with the Mann–Whitney U test when comparing the levels of a variable between two groups or with the Kruskal–Wallis test when there were more than two groups.

**Table 3 biomedicines-11-00225-t003:** Comparison of HPV L1 levels between selected groups.

Variable	L1 Level	MD (95% CI)	*p*
Histological outcome (with outliers)			
CIN 1 (n = 38)	1504.62 (154.65; 8683.80)	-	0.002
CIN 2 (n = 24)	3684.91 (90.45; 10,905.76)		
CIN 3 (n = 19)	13,566.25 (2099.17; 25,884.17)		
No pathology (n = 4)	25,123.51 (20,563.59; 33,437.23)		
Histological outcome (with outliers)			
CIN 1 (n = 38)	1504.62 (154.65; 8683.80) 3684.91 (90.45; 10,905.76)		0.022
CIN 2 (n = 24) CIN 3 (n = 19)	13,566.25 (2099.17; 25,884.17)		
Histological outcome (with outliers)			
CIN 1 + CIN 2 (n = 62)	1967.74 (133.21; 8713.67)	11,598.51 (949.47; 14,437.27)	0.006
CIN 3 (n = 19)	13,566.25 (2099.17; 25,884.17)		
16			
Yes (n = 40)	5107.77 (136.04; 17,349.82)	−685.53 (−4112.92; 795.55)	0.666
No (n = 45)	4422.24 (229.53; 13,566.25)		
18			
Yes (n = 5)	8713.67 (5009.37; 9210.31)	−4368.77 (−8457.52; 9937.55)	0.519
No (n = 80)	4344.90 (145.63; 14,213.32)	−685.53 (−4112.92; 795.55)	0.666
31			
Yes (n = 12)	1753.24 (351.93; 6430.07)	3256.13 (−1259.63; 6586.21)	0.579
No (n = 73)	5009.37 (154.65; 13,957.17)	−4368.77 (−8457.52; 9937.55)	0.519
45			
Yes (n = 4)	10,806.31 (4408.50; 17,007.96)	−6384.07 (−12,808.30; 8262.89)	0.514
No (n = 81)	4422.24 (154.65; 13,688.40)	3256.13 (−1259.63; 6586.21)	0.579
6			
Yes (n = 9)	593.55 (117.89; 13,566.25)	4514.22 (−2642.35; 6453.08)	0.424
No (n = 76)	5107.77 (192.09; 13,822.78)	−6384.07 (−12,808.30; 8262.89)	0.514
More than one HPV genotype			
Yes (n = 37)	5008.66 (402.65; 12,928.81)	−55.34 (−2553.91; 2034.14)	0.601
No (n = 48)	4953.32 (115.79; 17,093.67)	4514.22 (−2642.35; 6453.08)	0.424
Comorbidities			
Yes (n = 26)	5206.17 (1126.42; 14,469.47)	−3038.83 (−4752.51; 1344.42)	0.442
No (n = 59)	2167.34 (192.09; 13,247.53)	−55.34 (−2553.91; 2034.14)	0.601

CIN—cervical intraepithelial neoplasia; HPV—human papillomavirus. MD 95% CI—the median difference with the 95% confidence interval. Comparisons were made with the Mann–Whitney U test when comparing the levels of a variable between two groups or with the Kruskal–Wallis test when there were more than two groups.

## Data Availability

The data are available from the first author.

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
