# Peer review of "Expression of E4 Protein and HPV Major Capsid Protein (L1) as A Novel Combination in Squamous Intraepithelial Lesions"

_biomedicines, 2023, doi:10.3390/biomedicines11010225_

Round 1

Reviewer 1 Report

In the manuscript “E4 and HPV major capsid protein (L1) as a novel combination of biomarkers in cervical intraepithelial lesions” by Marcin et al. (Biomedicines-2099893), the authors aims to characterize the relationship between the presence of E4 and L1 HPV proteins in cervical biopsies and the relationship between the presence of these proteins and specific HPV genotypes in 85 women presenting abnormal liquid based cytology or Pap-smears, or presenting oncogenic HPV genotypes, or with clinical suspicion of cervical lesions. The authors performed immunohistochemistry to detect the presence of E4 and L1, and quantifying the presence using specific software. Levels of E4 and L1 were compared between women grouped in respect lesion severity, HPV status, and HPV genotype. Significant differences in levels of E4 were found between women grouped in respect to severity of cervical lesion, presence of HPV18 and HPV6; and in the levels of L1 for women grouped in respect to t severity of cervical lesion. The authors reported higher levels of E4 in CIN2, and higher levels of L1 in CIN3 and in HPV+ women without lesions, although these differences were not significant.

 The manuscript has some points that need to be clarified.

- The authors need to detail the procedure used for immunohistochemistry quantification of E4 and L1: how these values were obtained, negative and positive samples were used as references for these estimates?

- The authors need to clarify how samples with unknow HPV were characterized in respect to HPV genotype. How did these genotypes were identified?

- The authors need to inform the 37 genotypes identified and report if there were cases with multiple infections.

- The authors need to use a method to correct p-values as multiple comparisons were performed

Minor points:

- In line 267, substitute “Figures I-VII” by “Figures 1-7”.

- The citation Frederique et al. (line 320) was not listed in References.

-  Line 332, Zummeren et al., is the reference #23.

-  In discussion section lines 333-334 (“High-grade lesions (CIN 2/3) showed less E4 expression, which was inversely related to increasing hypermethylation.”, the authors need to clarify what is hypermethylated.

-In discussion section, lines 355-358, the authors need to provide references that support the statement.

Author Response

Dear Reviewer,

Thank you very much for your review and suggestions for improving the work, I would like to comment on them, point by point, and I believe that these corrections will make the work more valuable to the reader.

  1. The authors need to detail the procedure used for immunohistochemistry quantification of E4 and L1: how these values were obtained, negative and positive samples were used as references for these estimates?

Answer: Automatic detection of objects with a colour intensity greater by limit value enabled the identification of areas with a positive immunohistochemical reaction occurrence. The software detected the coloured area in the required intensity and recorded it as a numerical value in mm2 in the MS Exel file. The analyses included photos of the examined tissue fragments from patients who showed symptoms of HPV infection. Automatic classification and measurements were performed and used for further statistical analysis.

  1. The authors need to clarify how samples with unknow HPV were characterized in respect to HPV genotype. How did these genotypes were identified?

Answer: Informacja o genotypach HPV u pacjentek pochodziła z wymazów z szyjki macicy a następnie wykonania badań molekularnych, jak opisano w sekcji Materiały i Metody.

  1. The authors need to inform the 37 genotypes identified and report if there were cases with multiple infections.

Answer: Dodaliśmy informację o wykrywanych genotypach. The molecular test detected the DNA of 37 HPV genotypes to identify the following genotypes: 6, 11, 16, 18, 26, 31, 33, 35, 39, 40, 42, 43, 44, 45, 51, 52, 53, 54, 55, 56, 58, 59, 61, 62, 64, 66, 67, 68, 68a, 68b, 69, 70, 71, 72, 73, 81, 82, 83, 84, 87, CP6108, 90 in vitro.

  1. The authors need to use a method to correct p-values as multiple comparisons were performed.

Answer: The statistician working with us checked the calculations again. Can we have a more specific suggestion?

Minor points:

- In line 267, substitute “Figures I-VII” by “Figures 1-7”. à we’ve changed it

- The citation Frederique et al. (line 320) was not listed in References. à Thank You. We added it.

- Line 332, Zummeren et al., is the reference #23. à That’s deffinitelly right.

- In discussion section lines 333-334 (“High-grade lesions (CIN 2/3) showed less E4 expression, which was inversely related to increasing hypermethylation.”, the authors need to clarify what is hypermethylated. à We added the information to clarify discussion section. Promoter hypermethylation of host-cell genes involved in cervical carcinogenesis, is a marker for an advanced transforming HPV infection. During HPV-induced cervical carcinogenesis, the methylation levels increase with the severity of the underlying cervical disease and are exceptionally high in cervical cancer.

Reviewer 2 Report

Dear authors,

the study objective is important, addressing the need to improve patient management and follow up after the diagnosis of different grades of cervical dysplasia. Novel markers which can be used in disease outcome prediction are in scientific focus, so your study is valuable.

However, the manuscript could be improved, and suggestions follow:

1. Changing expressions/wording:

- suspicious cervix pictures to clinically suspicious cervix (in abstract and methods)

- cervical pathology to cervical lesion (in many places in the abstract and text)

2. Introduction

- the sentence in line 92 ("That is the formation of the....") is unclear - needs to be better explained

3. Materials and methods:

- part of the text in the "Study design" from line 119 to 134 belongs to introduction - I suggest moving it there

- part of the text in the "Study design" from line 135 to 137 describe the study objectives/aims. It is partly mentioned at the end of the Introduction. I propose that you summarize these two paragraphs in order to describe the aims of the study in one place.

- the last sentence in the "Study design" (lines 138-140, All the examinations....) can be moved up, in the line 118.

- for LBC - since you mention the results of cytology, you should describe the method used, I suppose it was ThinPrep

- HPV PCR method - it needs to be stated which method was used, was it commercial system or in-house

- immunohistochemistry - the subtitle 2.3.1. should start before paragraph beginning in line 167, text from line 167 to 182 need revision; description for L1 is unclear, different names of the producing company are stated, for E4 repeating of the company is not needed

- statistical analysis - in the line 202 I suppose it is meant to be "p value is set at 0,05" 

- a part of the results is shown with or without outliers, so, please describe the definition of the outliers, especially with so many in CIN1 category (14 out of 38)

- in the results you state the area in square micrometres for E4 and L1 - please describe the method used for area calculation (image analysis or other?)

4. Results

- in the line 243 "diagnosis" means histology? - it is not clear

- Figures - Figure 7 belongs to methods - immunohistochemistry, so it would become Figure 1

- figure 4, 5. and 6. are not necessary because the results were not significant

- figure 2 is not necessary, because it is similar to figure 3.

- remaining figures should be numerated accordingly

- in the description of the figures, after the word "levels", it should be added "of E4" (line 278), and (of L1) (line 282)

5. Discussion:

- include explanation of your results in regard of applying categories with and without outliers

- discuss more about L1 expression, because, in my opinion, L1 should have the highest expression in CIN1 and the lowest in CIN3, and your results were opposite. You can consult, discuss, and cite publications of dr Ralf Hilfrich group (on L1 in cytology). 

- at the end you could suggest the need of more future studies of HPV E4 and HPV L1, especially in cytological material, which could be informative before invasive procedures.

Author Response

Dear Reviewer,

Thank you very much for your review and suggestions for improving the work, I would like to comment on them, point by point, and I believe that these corrections will make the work more valuable to the reader.

  1. Changing expressions/wording:

- suspicious cervix pictures to clinically suspicious cervix (in abstract and methods) à We’ve changed it.

- cervical pathology to cervical lesion (in many places in the abstract and text) à We’ve changed it.

  1. Introduction

- the sentence in line 92 ("That is the formation of the....") is unclear - needs to be better explained  à This sentence was misunderstanding and should not be placed in this place. Thank You for your vigilance.

  1. Materials and methods:

- part of the text in the "Study design" from line 119 to 134 belongs to introduction - I suggest moving it there à That’s right. We moved it following your suggestion.

- part of the text in the "Study design" from line 135 to 137 describe the study objectives/aims. It is partly mentioned at the end of the Introduction. I propose that you summarize these two paragraphs in order to describe the aims of the study in one place. à That’s right. We moved it to the introduction and summarized it.

- the last sentence in the "Study design" (lines 138-140, All the examinations....) can be moved up, in the line 118. à That’s right. We moved it following your suggestion.

- for LBC - since you mention the results of cytology, you should describe the method used, I suppose it was ThinPrep à We were using the recommended and validated buffer- PreservCheck.

- HPV PCR method - it needs to be stated which method was used, was it commercial system or in-house à The method was commercial, but

- immunohistochemistry - the subtitle 2.3.1. should start before paragraph beginning in line 167, text from line 167 to 182 need revision; description for L1 is unclear, different names of the producing company are stated, for E4 repeating of the company is not needed à We changed it in case to be more clear.

- statistical analysis - in the line 202 I suppose it is meant to be "p value is set at 0,05" à  Thank you for this; we corrected the mistake.

- a part of the results is shown with or without outliers, so, please describe the definition of the outliers, especially with so many in CIN1 category (14 out of 38) à Finally, we resigned from the division into results with and without outliers. The differences concerned several people and did not have a statistically significant impact on the results of the analysis.

- in the results you state the area in square micrometres for E4 and L1 - please describe the method used for area calculation (image analysis or other?) à Automatic detection of objects with a colour intensity greater by limit value enabled the identification of areas with a positive immunohistochemical reaction occurrence. The software detected the coloured area in the required intensity and recorded it as a numerical value in mm2 in the MS Exel file. The analyses included photos of the examined tissue fragments from patients who showed symptoms of HPV infection. Automatic classification and measurements were performed and used for further statistical analysis

  1. Results

- in the line 243 "diagnosis" means histology? - it is not clear à We changed it into the phrase “histopathological outcome”.

- Figures - Figure 7 belongs to methods - immunohistochemistry, so it would become Figure 1 à We moved it according to your suggestion.

- figure 4, 5. and 6. are not necessary because the results were not significant à Yes, we agree that too many tables and scatter plots may let to losing the clue of the manuscript.

- figure 2 is not necessary, because it is similar to figure 3. à We agree, so we deleted it.

- remaining figures should be numerated accordingly à Now it is correct.

- in the description of the figures, after the word "levels", it should be added "of E4" (line 278), and (of L1) (line 282) à Now it is correct.

  1. Discussion:

- include explanation of your results in regard of applying categories with and without outliers à Finally, we resigned from the division into results with and without outliers. The differences concerned several people and did not have a statistically significant impact on the results of the analysis.

- discuss more about L1 expression, because, in my opinion, L1 should have the highest expression in CIN1 and the lowest in CIN3, and your results were opposite. You can consult, discuss, and cite publications of dr Ralf Hilfrich group (on L1 in cytology). à Thank You very much for suggesting us to read this interesting and valuable study. As Hilfrich et al. proves, the L1 capsid protein is detectable only at that stage of the life cycle. Therefore, detecting L1 capsid protein synthesized in the cells of the superficial layer of the epithelium might be easy to obtain while taking cervical swab.

- at the end you could suggest the need of more future studies of HPV E4 and HPV L1, especially in cytological material, which could be informative before invasive procedures. As it is really interesting point of view, we added few more sentences. We believe that it would be valuable for science and beneficial for patients in the future to develop diagnostic tests detecting L1 and E4 in cervical smears, which would significantly reduce the invasiveness of the test. However, it should be borne in mind that the cervical smear taken is superficial and we could obtain few L1 and E4-containing cells, while in tissue sections the assessment of layers and individual cells is much easier due to the amount of material.

Round 2

Reviewer 1 Report

There are some points that need to be clarified. Please, provide the answers in English.

- Material and Methods, item 2.2.1: This item is not clear. Please revise the text, it is not possible to understand how the HPV genotypes were identified. The reverse hybridization methodology needs to be better described. The authors mention that 37 different genotypes can be identified, however 40 genotypes are listed (6, 11, 16, 18, 26, 31, 33, 35, 39, 40, 42, 43, 44, 45, 51, 52, 53, 54, 55, 56, 58, 59, 61, 62, 64, 66, 67, 68, 68a, 68b, 69, 70, 71, 72, 73, 81, 82, 83, 84, 87, CP6108, 90). The authors mention that “The lab technicians performed sequence analysis to characterise HPV-positive samples with unknown HPV genotypes”. I understand that HPV positive samples where the genotype could not be identified by reverse hybridization, were submitted to DNA sequencing to genotype identification.

- The authors need to clarify if there were women with multiple infection.

-Lines 234-236: It is not possible to understand the text: “Figure 1 presents immunohistochemical expression of HPV in epithelial cells in the tissue of the same patients (clinical characterictics of patient A: CIN 1 in histopathological diagnosis, HPV 33, 62 (+) and patient B: CIN 2 in histopathological diagnosis, HPV 52 (+))”. The text information is not compatible with Figure 1 legend.

Author Response

Thank You very much for Your comments. Please read the attached file.

Round 3

Reviewer 1 Report

No additional comments.

Author Response

Thank You.